# 3D-RADNet: Extracting labels from DICOM metadata for training general medical domain deep 3D convolution neural networks

**Richard Du**[1]                                                                    DU94@HKU.HK

**Varut Vardhanabhuti**[1]                                                           VARV@HKU.HK

[1] *Department of Diagnostic Radiology, Li Ka Shing Faculty of Medicine, The University of Hong Kong, Hong Kong SAR*

## Abstract

Training deep convolution neural network requires a large amount of data to obtain good performance and generalisable results. Transfer learning approaches from datasets such as ImageNet had become important in increasing accuracy and lowering training samples required. However, as of now, there has not been a popular dataset for training 3D volumetric medical images. This is mainly due to the time and expert knowledge required to accurately annotate medical images. In this study, we present a method in extracting labels from DICOM metadata that information on the appearance of the scans to train a medical domain 3D convolution neural network. The labels include imaging modalities and sequences, patient orientation and view, presence of contrast agent, scan target and coverage, and slice spacing. We applied our method and extracted labels from a large amount of cancer imaging dataset from TCIA to train a medical domain 3D deep convolution neural network. We evaluated the effectiveness of using our proposed network in transfer learning a liver segmentation task and found that our network achieved superior segmentation performance (DICE=90.0%) compared to training from scratch (DICE=41.8%). Our proposed network shows promising results to be used as a backbone network for transfer learning to another task. Our approach along with the utilising our network, can potentially be used to extract features from large-scale unlabelled DICOM datasets.

**Keywords:** Transfer learning, Large dataset, Data mining

## 1. Introduction

With the abundance of medical images routinely taken at hospitals, imaging-based machine learning approaches had become a centre of development in diagnostic radiology. Methods based on deep learning had promising results in many areas of diagnostic radiology such as tumour segmentation and classification (Gonzalez et al., 2018; Song et al., 2018; Nielsen et al., 2018). Despite successes, performances of neural networks in the medical domain are often limited by small development set. To alleviate the problem, transfer learning from pre-existing datasets were used to improve performance. Transfer learning is the process of taking existing pre-trained network architecture designed for an existing dataset (typically over a million) and then fine-tuned against on data for another task. The current most popular dataset used for transfer learning is ImageNet, which consists of over 14 million natural images with 20 thousand classes (Russakovsky et al., 2014). Popular ImageNet architec-

tures such as ResNet, Inception and DenseNet had had success in medical imaging task, particular in radiographs and ophthalmological images such as a retina scan (Rajpurkar et al., 2017; Gulshan et al., 2016). However, since most diagnostic imagery consists of 3D volumetric images, transfer learning from the natural domain are not feasible. Since 3D networks architecture generally is more prone to over-fitting due to having more parameters, this had often lead to some studies to treat each 2D slices of a 3D volume independently to leverage transfer learning, hence discarding the potential useful structural information.

Despite many radiological imaging datasets are available in the public domain from channels such as CodaLab (https://competitions.codalab.org) and The Cancer Imaging Archive (TCIA) (Clark et al., 2013), there has not been a large-scale 3D dataset that is available to use for training a network similar to ImageNet. One primary reason is that medical images are inherently more complex than natural images, and would require a significant amount of time and specialised medical knowledge to annotate. The recent effort by Chen et al. (2019) had developed a general multi-domain network (MED3D) based on publicly available volumetric segmentation datasets had shown superior performance in several organ segmentation tasks compared to training from scratch, further emphasises the need for a large-scale dataset. One limiting factor of MED3D is that it requires segmentation annotations which could be impractical on a large scale due to the time needed for annotating volumetric images. An alternative method is needed to build such network and dataset for transfer learning in the medical domain.

One potential avenue to explore is the use of digital imaging and communications in medicine metadata (DICOM). DICOM is the standardised format for storing and transferring medical images in clinics. Along with the imagery, DICOM metadata stores patient information and acquisition parameters of the scan. For example, information on the type of imaging modality, types of MRI sequences used, patient position during the scan, and the use of contrast agent could potentially provide enough distinct features to describe the appearance of the images for a neural network to learn. In this study, we explore whether we can automatically or semi-automatically extract labels from DICOM metadata of a large amount of DICOM images from publicly available datasets to train a general medical domain convolution neural network.

The main contributions of this study are as follows[1]:

- We acquired and semi-automatically labelled a large public MRI and CT dataset available from TCIA by using the information provided in the DICOM headers.

- We trained a 3D convolution neural network on a large amount of volumetric radiological scans to classify modality, imaging sequence, view, presence of contrast agent, and the coverage of the body part.

- We demonstrated the effectiveness of using our proposed network for transfer learning of a liver segmentation task. We found a high performance gained compared to training a 3D convolution neural network from scratch.

---

1. To facilitate future development and application of our network and data. Source code, dataset and labels will be made publicly at https://github.com/du1388/3d-radnet

## 2. Methodology

This study aims to develop and train a general medical domain network from a large amount of 3D volumetric data that could be used for transfer learning to other medical imaging tasks. To achieve this, we acquired and analysed a large collection of cancer imaging data from the TCIA database to extract useful labels to train a convolution neural network called the 3D-RADNet. We then tested the effectiveness of the network in transfer learning by using the network as an encoder for segmentation of the liver.

### 2.1. Data Acquisition and Label Extraction

TCIA is an online database that hosts a large number of medical images of cancer. All images are in DICOM format and organised into different collections based on the type of diseases. For this study, we downloaded all collections that contain MRI and CT scans and can be redistributed under the creative commons attribution 3.0 unsupported license (https://creativecommons.org/licenses/by/3.0/. A list of all the TCIA collections acquired for this study are given in Appendix A Tables A1 and A2. Once all the scans were acquired, we extracted all the standardised DICOM metadata from all scan series in the collections for analysis.

#### 2.1.1. MRI Sequences

The appearance of an MRI image is dependent on the MRI sequences used for the acquisition. Commonly used diagnostic sequences can be classified into three types: spin-echo (SE), inversion recovery (IR) and gradient-echo (GR). This is specified in the DICOM attributes *Scanning Sequence* (0018,0020) under the same classification. Two main SE sequences, T1-weighted (T1 - SE) and T2-weighted (T2 - SE) are commonly used in diagnostic scans. The differences between weighing can be determined by the attributes *Repetition time* (0018,0080) and *Echo time* (0018,0081), where a T1-weighted scan have a short repetition time (TS) and short echo time (TE), and vice versa. For IR sequences, fluid-attenuated inversion recovery (FLAIR) and short tau inversion recovery (STIR) are most commonly used. The weighting of IR sequences can be determined by the TE time of the sequence. In addition to scanning sequence and TS/TE times, series (0008,103E) and study description (0008,1030) can also be used to identify types of sequences. However the descriptions are not standardised and can vary greatly depending on the convention used by the imaging centre and vendor. Due to a high amount of different variants and different name conventions of GR sequences, it is difficult to group the sequences accordingly. Therefore We decided not to use GR in our analysis. Other types of functional imaging sequences such as functional MRI, magnetic resonance angiography, diffusion and perfusion-weighted imaging was also excluded in our study due to vast differences in appearances. Time-series images such as dynamic-contrast were also excluded to avoid biases as there will be a high amount of the same scan present in training.

#### 2.1.2. Scan view

The anatomically plane in which the scan was taken can be identified by the attributes *Patient Image Position* (0020,0032) and *Patient Image Orientation* (0020,0037). Patient

image position specifies the (x,y,z) coordinates of the upper left-hand corner of the image, whereas image orientation describes the direction cosine of the first row and the first column with respect to the patient. For most standard patient orientation of scans,a image orientation of [1,0,0,0,1,0], [1,0,0,0,0,-1], [0,1,0,0,0,-1] corresponds to axial, coronal and sagittal view respectively. For non-standard orientations, the view can be determined by *Patient position* (0018,5100) attributes. However, the occurrence of a non-standard view is rare. Hence we excluded all the cases from our analysis.

### 2.1.3. Contrast agent

The presence of a contrast agent in MRI and CT imaging can significantly affect the appearance of the image. The use of contrast agent are recorded in the attributes *Contrast/Bolus Agent* (0018,0010).

### 2.1.4. Scan coverage label

The scan coverage of the body was explored to provide structural information of the image to the network. As scans protocols are often standardised in practice, extracting the target and coverage of the scan can systematically be obtained by comparing the study and series description, type of cancer given by TCIA and size of the scan. The label scheme for the coverage is shown in Figure 1. For each body parts/organs, the scan must cover the entirety of the target to be considered. For upper head and neck, it must include the sphenoid sinus, nasopharynx and oropharynx. Lower head and neck, it must cover from the larynx to apex of the lung.

### 2.1.5. Image Processing

All scans with less than 16 slices were excluded from the study to ensure there are sufficient slices for the network. To address heterogeneous voxels sizes and slice spacing across the scans, all scans were linearly resized to 48x192x192, which is the input size of the network. For scans with less than 48 slices, the scans were centred and filled with blank slices up to 48 slices. The effective slice spacing after resizing was recorded, and all resized scans with spacing higher than 1.5cm were excluded. All scans were then normalised by min-max normalisation and discretised to 256 grey levels.

## 2.2. 3D-RADNet Network

The proposed 3D-RADNet takes an input image of 48x192x192 and outputs five outputs classifying the image modality/sequences, view, contrast, scan coverage and slice spacing. For the network architecture, we adapted a ResNet50 structure to take 3D inputs (He et al., 2015). The network then connected to a fully connected layer of 1000 neurons and then branches into five separate layers corresponding to each of the outputs. For modality/sequence, view and contrast, softmax activation was used. Sigmoid activations were used of the scan coverage layer, and linear activation was used for slice spacing. The parameters of the network were optimised using cross-entropy loss and root-mean-squared with by ADAM optimiser.

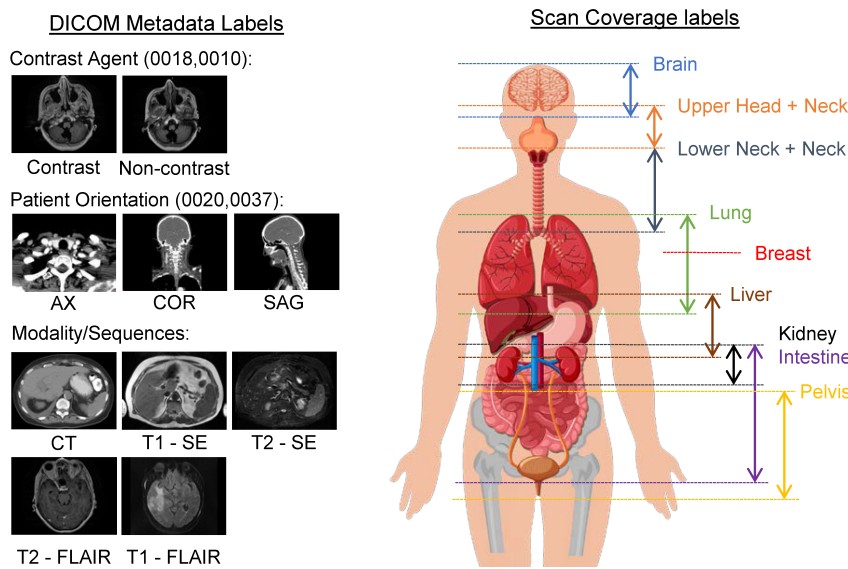

Figure 1: Scan coverage of labels used, and extracted labels from the DICOM metadata.

## 2.3. Liver segmentation

To evaluate the effectiveness of transfer learning in the proposed 3D-RADNet, we applied the network to segment the liver in the LiTS challenge dataset (https://competitions.codalab.org/competitions/17094. The LiTS challenge dataset consists of 131 abdominal CT scans with detailed delineation of the organ. Due to different slice spacing across the scans, all scans were linearly interpolated into 5mm spacing. The framework for transfer learning is described in Figure 2. The framework consists of two steps, the liver region proposal, and the training of the segmentation network. A sliding window of 48 slices with a stride of 8 slices was applied to extract patches of the potential region covering the liver. The extracted areas are evaluated with the 3D-RADNet to identify the regions containing the liver. As multiple patches from the sliding window can arise from one case, a threshold of 0.9 was used as cut off for selection. If no patches were greater than 0.9, the maximum score was used. The selected regions were then used for subsequent training. The decoder for the segmentation network was based on the VNet structure with up-sampling and skip-connection (Milletari et al., 2016). The parameters of the network were optimised using dice loss with by ADAM optimiser. To determine the impact of modification of the 3D-RADNet on the performance of segmentation, we trained different networks by freezing different residual blocks of the network. In addition, we also evaluate the impact of training sample in 3D-RADNet for transfer learning liver segmentation by training the network with random subsets of the training data.

## 3. Results

Of the 57 datasets acquired from TCIA, a total of 63276 (6544 unique subjects) individual MRI and CT image series were analysed. 45609 series were excluded from the study (in-

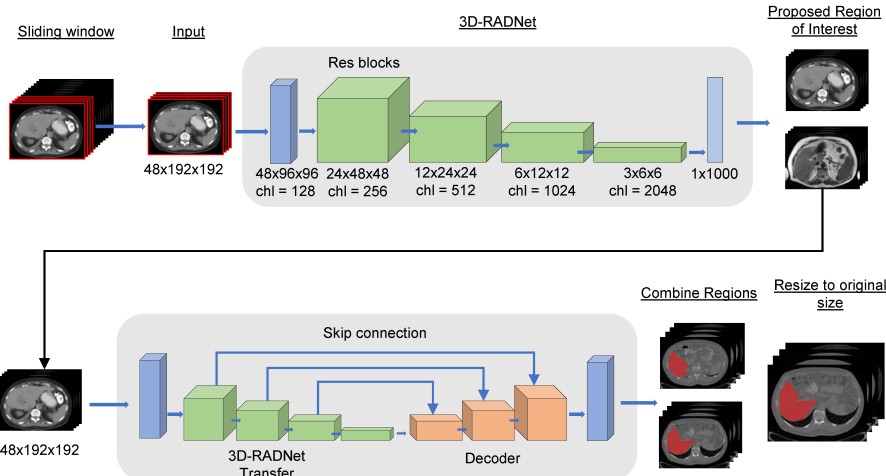

Figure 2: Transfer learning framework for liver segmentation task.

consistent/invalid slice position or spacing: 14575; the number of slices less than 16: 3768; missing series descriptions: 2041; excluded MRI sequences: 10887; Time-series: 14338). STIR sequence was also excluded from the study because of a low number of scans found. The remaining 17667 (4453 unique subjects) series were split into a training, validation and testing set for training the network. The summary of the data is shown in Table A3.

### 3.1. 3D-RADNet

The classification results of the 3D-RADNet are summarised in Table 1. Of all the classifications, the presence of contrast agent had the worse accuracy (84.8%) with all other classification achieving over 90% in the testing set. A mean absolute error of 3.1mm and root mean square error of 4.2mm was obtained in the testing set for slice spacing regression.

### 3.2. Transfer Learning and Liver Segmentation

For the training of the segmentation network, the dataset was split 70:15:15 into training (n=92), validation (n=19) and testing set (n=20). Table 2 shows the impact of performance by the degree of modification of the 3D-RADNet for liver segmentation. The best segmentation performance was obtained from the network trained with all encoding 3D-RADNet layers frozen (DICE = 90.0%). Retraining the encoding res-block 3 and 4 achieved the second-highest score (DICE = 84.3%). Lowest segmentation performance was seen from training the network from scratch and from initialised with 3D-RADNet weights without freezing any layers. This was further illustrated in Appendix A Figure A1 where it can be seen that the networks failed to generalised to the validation set. For the best performing network, the impact of reducing training samples on the segmentation performance was analysed. As expected, Table 3 shows that performance decrease with training samples

with a Dice score of 80% achieved with 20% training data (n=18). For all networks, the validation set was kept the same.

Table 1: Classification performance of 3D-RADNet on the testing set

|  | Samples | Accuracy | AUC | Recall | Precision |
|---|---|---|---|---|---|
| **Modality/Sequence** | | | | | |
| CT | 117 | 100% | 100% | 100% | 100% |
| T1 - SE | 51 | 97.8% | 98.1% | 92.2% | 94.0% |
| T2 - SE | 101 | 96.8% | 98.8% | 99.0% | 91.7% |
| T1 - FLAIR | 27 | 99.7% | 100% | 100% | 96.4% |
| T2 - FLAIR | 20 | 97.5% | 96.5% | 60% | 100% |
| Total | 316 | 95.9% | | | |
| **View** | | | | | |
| Axial | 267 | 99.4% | 100.0% | 99.3% | 100% |
| Coronal | 32 | 100% | 100% | 100% | 100% |
| Sagittal | 17 | 99.4% | 100.0% | 100% | 89.5% |
| Total | 316 | 99.4% | | | |
| **Contrast** | | | | | |
| Contrast | 100 | 84.8% | 91.7% | 86.0% | 71.7% |
| No Contrast | 216 | 84.8% | 91.7% | 84.3% | 92.9% |
| **Scan Coverage** | | | | | |
| Brain | 200 | 98.4% | 99.9% | 97.5% | 100% |
| Upper Head-Neck | 42 | 98.7% | 99.7% | 90.5% | 100% |
| Lower Head-Neck | 38 | 98.7% | 98.7% | 89.5% | 100% |
| Lung | 64 | 98.4% | 98.8% | 93.8% | 98.4% |
| Breast | 82 | 98.7% | 99.6% | 96.3% | 98.8% |
| Liver | 52 | 99.1% | 99.7% | 98.1% | 96.2% |
| Kidney | 52 | 98.7% | 99.8% | 100% | 92.9% |
| Intestine | 46 | 98.1% | 100.0% | 89.1% | 97.6% |
| Pelvis | 55 | 98.1% | 99.9% | 96.4% | 93.0% |
| Total | 316 | 91.5% | | | |

## 4. Discussion

In this study, we developed and trained a medical-domain 3D convolution neural network that can be used as a backbone network for transfer learning. We achieved this by acquiring a large amount of cancer imaging dataset from TCIA. We devised a scheme to extract labels from DICOM metadata to train the network. The labels extracted include imaging modalities and sequences, view, presence of contrast agent, slice spacing and scan coverage. Of all the labels, extracting the scan coverage was the most time-consuming part. As datasets from the TCIA came from a range of different centres and institutes, it was difficult to learn the structure of the cases. However, in practice, this would not be a problem when applied to a single centre as you would know the scanning protocols beforehand. To

Table 2: Impact of liver segmentation performance by freezing different layers of the network and comparison of performance to no transfer learning is also given.

| Modification | Mean DICE | Mean IOU | Median DICE | Median IOU |
|---|---|---|---|---|
| Froze all layers | 90.0% | 80.7% | 90.0% | 81.8% |
| Froze to block 3 | 72.1% | 59.7% | 81.0% | 68.1% |
| Froze to block 2 | 84.3% | 76.8% | 90.3% | 82.3% |
| Froze to block 1 | 77.6% | 65.1% | 81.8% | 69.3% |
| Weights only | 44.8% | 38.1% | 43.0% | 27.4% |
| No transfer learning | 41.8% | 38.0% | 43.0% | 27.3% |

Table 3: Impact of liver segmentation performance by percentage of training samples

| % Training Data | Mean DICE | Mean IOU | Median DICE | Median IOU |
|---|---|---|---|---|
| 100% (n=92) | 90.0% | 81.6% | 90.1% | 82.0% |
| 80% (n=74) | 84.7% | 75.0% | 89.0% | 80.0% |
| 60% (n=55) | 84.2% | 73.9% | 87.2% | 77.3% |
| 40% (n=37) | 87.0% | 77.9% | 90.9% | 83.4% |
| 20% (n=18) | 80.0% | 69.30% | 86.5% | 76.1% |

evaluate the effectiveness of our network for transfer learning, we applied the network to a liver segmentation task. The results show that transfer learning by using our network achieved significantly higher performance than training from scratch in 3D networks. This is expected as the higher number of variables in 3D compared to 2D networks means would be more prone to over-fitting. This was also reflected in that the network only generalised well when the layers of the network were fixed regardless of transfer learning.

In comparison to earlier work by Chen et al. (2019), our network achieved a lower but comparable performance (DICE=90.0%) compared to MED3D (DICE=94.6%) that was trained on segmentation datasets. One difference in our approach was that we did not refine the segmentation mask from the initial segmentation. As we wanted to evaluate the raw performance of segmentation from the whole image. We would expect the performance to increase by refining the mask from expanding the region by using the initial segmentation mask.

One major limitation of our work is that the network architecture used for the 3D-RADNet is a 3D adaptation of ResNet50 which developed for 2D natural images. ResNet may not be optimal for volumetric medical images. Alternative network architecture tailed for medical images may increase the performance of the network.

In conclusion, we presented an approach to extract labels from DICOM metadata that describes the appearance of the images to train a general medical domain network. The 3D-RADNet shows promising potential to be used as a backbone network for transfer learning to another task. Our approach, along with the network, can potentially be used to extract features from large-scale unlabelled DICOM datasets.

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

## Appendix A.

Table A1: List of TCIA dataset used for the training of the proposed 3D-RADNet network

| TCIA Sets | Modality | Location | Patient |
|---|---|---|---|
| AAPM RT-MAC (Cardenas et al., 2019) | MRI | Head-Neck | 55 |
| Brain-Tumor-Progression (Schmainda and Prah, 2018) | MRI | Brain | 20 |
| C4KC-KiTS (Heller et al., 2019) | CT | Kidney | 210 |
| Breast-MRI-NACT-Pilot (Newitt and Hylton, 2016) | MRI | Breast | 64 |
| CPTAC-CCRCC (CPTAC, 2018a) | MRI, CT | Kidney | 63 |
| CPTAC-GBM (CPTAC, 2018c) | MRI, CT | Brain | 63 |
| CPTAC-HNSCC (CPTAC, 2018d) | MRI, CT | Head-Neck | 55 |
| CPTAC-LUAD (CPTAC, 2018e) | MRI, CT | Lung | 32 |
| CPTAC-PDA (CPTAC, 2018f) | MRI, CT | Pancreas | 68 |
| CPTAC-UCEC (CPTAC, 2018h) | MRI, CT | Uterus | 60 |
| CT COLONOGRAPHY (Smith et al., 2015) | CT | Colon | 825 |
| Head-Neck Cetuximab (Bosch et al., 2015) | CT | Head-Neck | 111 |
| ISPY1 (Newitt and Nola, 2016) | MRI | Breast | 222 |
| IvyGAP (Nameeta et al., 2016) | MRI, CT | Brain | 39 |
| LGG-1p19qDeletion (Erickson et al., 2017) | MRI | Brain | 159 |
| LungCT-Diagnosis (Grove et al., 2015) | CT | Lung | 61 |
| Lung-Fused-CT-Pathology (Rusu et al., 2017) | CT | Lung | 6 |
| NSCLC-Radiomics-Genomics (Aerts et al., 2014) | CT | Lung | 89 |
| Pancreas-CT (Roth et al., 2015) | CT | Pancreas | 82 |
| Pelvic-Reference-Data (Yorke et al., 2019) | CT | Pelvis | 58 |
| Prostate Fused-MRI-Pathology (Madabhushi and Feldman, 2016) | MRI | Prostate | 28 |
| PROSTATE-DIAGNOSIS (Bloch et al., 2015b) | MRI | Prostate | 92 |
| PROSTATE-MRI (Choyke et al., 2016) | MRI | Prostate | 26 |
| PROSTATEx (Litjens et al., 2014) | MRI | Prostate | 346 |
| QIN Breast DCE-MRI (Huang et al., 2014) | MRI | Breast | 10 |
| QIN LUNG CT (Kalpathy-Cramer et al., 2016) | CT | Lung | 47 |
| REMBRANDT (Scarpace et al., 2015) | MRI | Brain | 130 |
| RIDER Lung CT (Zhao et al., 2009) | CT | Lung | 32 |
| SPIE-AAPM Lung CT Challenge (Armato et al., 2015) | CT | Lung | 70 |
| TCGA-BLCA (Kirk et al., 2016b) | MRI, CT | Bladder | 120 |
| TCGA-BRCA (Lingle et al., 2016) | MR | Breast | 139 |

Table A1 continues: List of TCIA dataset used for the training of the proposed 3D-RADNet network

| TCIA Sets | Modality | Location | Patient |
|---|---|---|---|
| TCGA-CESC (Lucchesi and Aredes, 2016a) | MR | Cervix | 54 |
| TCGA-COAD (Kirk et al., 2016e) | CT | Colon | 25 |
| TCGA-ESCA (Lucchesi and Aredes, 2016b) | CT | Esophagus | 16 |
| TCGA-GBM (Scarpace et al., 2016) | MRI, CT | Brain | 262 |
| TCGA-HNSC (Zuley et al., 2016) | MRI, CT | Head-Neck | 227 |
| TCGA-KICH (Linehan et al., 2016a) | MRI, CT | Kidney | 15 |
| TCGA-KIRC (Akin et al., 2016) | MRI, CT | Kidney | 267 |
| TCGA-KIRP (Linehan et al., 2016b) | MRI, CT | Kidney | 33 |
| TCGA-LIHC (Erickson et al., 2016a) | MRI, CT | Liver | 97 |
| TCGA-LUAD (Albertina et al., 2016) | CT | Lung | 69 |
| TCGA-LUSC (Kirk et al., 2016a) | CT | Lung | 37 |
| TCGA-OV (Holback et al., 2016) | MRI, CT | Ovary | 143 |
| TCGA-STAD (Lucchesi and Aredes, 2016c) | CT | Stomach | 46 |
| TCGA-UCEC (Erickson et al., 2016b) | MRI, CT | Uterus | 65 |

Table A2: List of TCIA dataset used for the testing of the proposed 3D-RADNet network

| TCIA Sets | Modality | Location | Patient |
|---|---|---|---|
| Anti-PD-1 MELANOMA (Patnana et al., 2019) | MRI, CT | Skin | 47 |
| BREAST-DIAGNOSIS (Bloch et al., 2015a) | MRI, CT | Breast | 88 |
| CPTAC-CM (CPTAC, 2018b) | MRI, CT | Skin | 92 |
| CPTAC-SAR (CPTAC, 2018g) | MRI, CT | Extremities | 22 |
| HNSCC (CPTAC, 2018d) | MRI, CT | Head-Neck | 55 |
| QIN-BRAIN-DSC-MRI (Schmainda et al., 2016) | MRI | Brain | 49 |
| Soft-tissue-Sarcoma (Vallières et al., 2015) | MRI | Skin | 51 |
| TCGA-LGG (Pedano et al., 2016) | MRI, CT | Brain | 199 |
| TCGA-PRAD (Zuley et al., 2016) | MRI, CT | Prostate | 14 |
| TCGA-READ (Kirk et al., 2016d) | MRI, CT | Kidney | 3 |
| TCGA-SARC (Roche et al., 2016) | MRI, CT | Extremities | 5 |
| TCGA-THCA (Kirk et al., 2016c) | CT | Neck | 6 |

Table A3: Breakdown of the training and testing set

| | Training set | Validation set | Testing set |
|---|---|---|---|
| | n=12000 | n=3305 | n=316 |
| **Modality/Sequence** | | | |
| CT | 5149 | 1556 | 117 |
| T1 - SE | 2187 | 652 | 51 |
| T2 - SE | 3477 | 1017 | 101 |
| T1 - FLAIR | 463 | 207 | 27 |
| T2 - FLAIR | 724 | 273 | 20 |
| **View** | | | |
| Axial | 8424 | 2202 | 267 |
| Coronal | 1298 | 426 | 32 |
| Sagittal | 2278 | 677 | 17 |
| **Contrast** | | | |
| Contrast | 8619 | 2407 | 216 |
| No Contrast | 3381 | 898 | 100 |
| **Scan Coverage** | | | |
| Brain | 4966 | 1677 | 200 |
| Upper Head-Neck | 1070 | 266 | 42 |
| Lower Head-Neck | 953 | 191 | 38 |
| Lung | 1002 | 271 | 64 |
| Breast | 1011 | 303 | 82 |
| Liver | 2818 | 693 | 52 |
| Kidney | 3544 | 846 | 52 |
| Intestine | 2179 | 504 | 46 |
| Pelvis | 3901 | 845 | 55 |

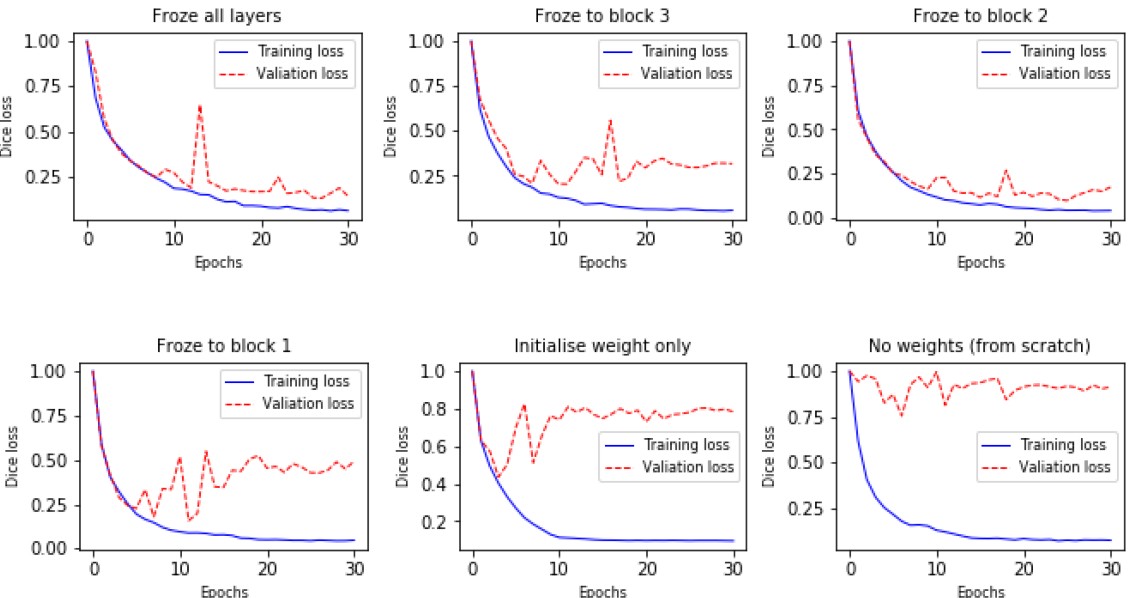

Figure A1: Training and validation losses during training

