# OpenReview forum: "3D-RADNet: Extracting labels from DICOM metadata for training general medical domain deep 3D convolution neural networks"
_MIDL.io/2020/Conference — MIDL 2020_

### Official Review · AnonReviewer2 · 2020-03-10
**Automatically using DICOM metadata to build training cohort for 3D CNNs**

**Rating:** 3
**Confidence:** 5

**Summary:**

- Built a 3D network to identify relevant images for training based on identifying different image characteristics (view, resolution, contrast)
- Ran on TCIA to build a cohort for training a segmentation network to classify DICOM tags based on input images
- Application to liver segmentation on a reasonably sized cohort.

**Strengths:**

- Well motivated introduction explaining the technical need for building medical image specific networks and training cohorts.
- Specific approach here is interesting, but unclear how the different networks come together
- Good summary of methods and explanation of how each DICOM tag can be used
- Fairly comprehensive evaluation of both networks, even if segmentation performance is on a small cohort

**Weaknesses:**

- A little unclear how the DICOM-based network is to be used. Once you identify relevant scans they still need annotations to be used for training? Is the goal to have as many scans identified as possible?
- Not clear why we are predicting DICOM tags using images.. is this in the case that the tag is not part of the image?
- Not clear why 3D-RADNet is being applied to a liver segmentation dataset.
- No statistical testing performed

**Justification Of Rating:**

Interesting ideas and reasonable validation on 2 different cohorts. But exact use of network and experimental design are very unclear. Some limitations mentioned by authors fundamentally undermine the paper.

**Paper Type:**

both

**Questions To Address In The Rebuttal:**

- Clarify the experimental design
- Run statistical testing to confirm significance of results

**Special Issue:**

no

---

> ### Author Response · Authors · 2020-03-27
> **Response to review 2**
>
> We would like to thank reviewer 2 for reading and reviewing our work and providing insightful comments and suggestions.
>
> We would like to clarify and address the following points:
>
> -	“A little unclear how the DICOM-based network is to be used. Once you identify relevant scans, they still need annotations to be used for training? Is the goal to have as many scans identified as possible?”
>
> Thank you for the comments. We apologise for not making this more clearly. The goal of our study is to find a way to easily provide labels to as much scans as possible. To do this we used the DICOM attributes in the DICOM headers to provide information on the scan to generate the labels that could used to train a CNN. As you have pointed out, for the body coverage label, we would still need to annotate it. For the TCIA datasets, we utilised the dataset information along with the DICOM study and series descriptions to sort the labels. For example, a head-neck cancer dataset would mostly consist of head and neck scans. However, occasionally you would find a chest CT or full body CT scans in a head-neck dataset for metastasis detection purposes. These scans can be easily picked up by using the study descriptions alone or by comparing the size of the scan. Once we remove those scans, we manual sampled and inspected a few scans to weakly label the body coverage for rest of the scans in the dataset. This labelling step was the most time consuming for the TCIA dataset as there were a lot of different datasets from different centres using different scanners. Also, we did not have any prior knowledge of the imaging protocols used by these centres. Although we haven’t tried it, we believed to prospectively apply the label methods on new data from a single centre would be easier as you would know the imaging protocols used and also the scans procedure will be more standardised in one centre.
> We will clarify this better in the revised submission.
>
> -	“Not clear why we are predicting DICOM tags using images. is this in the case that the tag is not part of the image?”
>
> We apologise for the misunderstanding. As DICOM tags are normally available, our purpose was not to predict DICOM tags. The goal of our study is to see whether we can use DICOM tags to extract some useful labels that provides information on the features and appearance of the image to train a CNN that would be used in the the medical domain for transfer learning applications.
> Although not a main focus, we have shown that we could use our model to automatically localise an image to a specific body region by using a sliding window approach as demonstrated in the segmentation experiments.
>
> -	“Not clear why 3D-RADNet is being applied to a liver segmentation dataset.”
> Sorry for not being clear with the methods. The purpose of the liver segmentation experiments in our study was to demonstrate the performance of transfer learning using the 3D-RADNet.
>
> -	“No statistical testing performed”
>
> Thank you for the suggestion. For the 3D-RADNet testing results in step one, we did not compare the results to another model. For the segmentation experiments, we compared the results of models trained with different training size and different degree of transfer learning by freezing the layers. However, after researching on this matter, we struggled to find a hypothesis/statistical test for segmentation results therefore we only presented the DICE coefficient and Intersection-Over-Union IOU results. We would greatly appreciate, if the reviewer could point us to an appropriate test for testing segmentation results.
>
> -	“Some limitations mentioned by authors fundamentally undermine the paper.”
>
> As mentioned in the manuscript, we agree that the current network architecture may not be optimal for volumetric images. However, the main purpose of our study was to validate whether our approach in generating labels to train CNN works and that its useful for transfer learning. We agree that finding an optimal network architecture for 3D images is an important step, but we it would a subsequent study to investigate. We also would like to point out that we have repackaged and release the dataset used in the training of our model to encourage and facilitate others to train and test their own architecture for their specific tasks.

---

### Official Review · AnonReviewer1 · 2020-03-13
**ImageNet for medical images**

**Rating:** 4
**Confidence:** 4
**Recommendation:** Oral

**Summary:**

This is a very interesting paper that demonstrates a strategy for building an "ImageNet" for medical imaging that can be used for transfer learning. The idea is to train a network to classify DICOM attributes, which are easy to automatically extract. They then show that transfer learning applied to a specific tasks improves performance compared to training from scratch.

**Strengths:**

This is a great idea for how to generate a pre-trained model for use with general deep learning based medical image analysis. The idea is simple yet powerful, and the fact that the concept was tested on a common deep learning based medical image analysis problem - segmentation - makes this overall a great paper.

**Weaknesses:**

It isn't clear why freezing all of the layers in the network would lead to a better segmentation performance. A more detailed discussion of this, or at least a reference to a paper that discusses this topic, would be beneficial.

It would have been optimal if there was an experiment added that compared transfer learning performance using the DICOM tag labeling network vs a network trained on ImageNet.

**Justification Of Rating:**

A high rating is given to this paper because it is likely to have a high impact.  Medical imaging doesn't have it's own "ImageNet" that can be used for transfer learning from the domain of medical images. This paper shows how to do that and  shows that it works well. The experiments are well designed and the results are impressive.

**Paper Type:**

both

**Special Issue:**

yes

---

> ### Author Response · Authors · 2020-03-27
> **Reponse to Review 1**
>
> We would like to thank reviewer 1 for reading and reviewing our work and providing insightful comments and suggestions.
>
> We would like to clarify and address the following points:
>
> -	“It isn't clear why freezing all of the layers in the network would lead to a better segmentation performance. A more detailed discussion of this, or at least a reference to a paper that discusses this topic, would be beneficial.”
>
> Thank you for the suggestion. This is one observation we neglected to discuss in more details in the paper. We believe that given the large 3D input size of the network and the small sample size of the lung segmentation data, the network is more prone to overfitting. This was reflected in Figure A1 where the validation loss is high when you allow all the parameters of the network to train. By freezing the network, we will force the segmentation network to decode the features extracted by 3D – Radnet only. We will address this more in the discussion of the revised submission.
>
> -	“It would have been optimal if there was an experiment added that compared transfer learning performance using the DICOM tag labeling network vs a network trained on ImageNet.”
>
> Thank you for the suggestion. We also believe that it would be good validation to compare it to a 2D segmentation method. However, due to the virus pandemic in Hong Kong, we are unable to access the GPU workstation to conduct any more experiments. This is one area we plan to work on and we hope to release the results as soon as possible either in the paper/presentation or on the GitHub page.

---

### Official Review · AnonReviewer4 · 2020-03-15
**An attempt to create an equivalent of ImageNet in the medical imaging space using TCIA radiology cohorts**

**Rating:** 3
**Confidence:** 4
**Recommendation:** Poster

**Summary:**

A well written paper, and a great hypothesis that can be used as a baseline for several application based projects in the medical imaging community. The authors extracted labels from a large amount of cancer imaging dataset from TCIA to train a medical domain 3D deep convolution neural network.Then, they evaluated the effectiveness of their proposed network in transfer learning a liver segmentation task and report that their network achieved superior segmentation performance (DICE=90.0%) compared to training from scratch (DICE=41.8%).

**Strengths:**

* A well written paper with all the information provided in the Supplementary as well.
* Good use of diagrams
* Good 'n' in the dataset curation
* Well laid out hypothesis
* Technically sound
* Code has been shared for reproducibility via github

**Weaknesses:**

There a few minor typos in the submitted paper. I suggest the authors to give the paper a thorough read.
For somebody with no previous knowledge, it is suggested that the authors spend some time and explain what DICOM
attributes mean. For example, if  Scanning Sequence is known by (0018,0020)-- is this universal and consistent across all sites? or is this vendor specific?

**Justification Of Rating:**

The authors have attempted to establish an equivalent of ImageNet in the radiology space by using multiple TCIA cancer cohorts. Massive datasets have been curated and QCed for this ourpose. The paper should be recognised for this purpose. The authors have also gone one step further by demonstrating the use of such a model in applications by segmentation the liver.

**Paper Type:**

both

**Questions To Address In The Rebuttal:**

* Have the authors made this trained model publicly available for immediate use?
* If it is not too much effort, I would also like to see another application implementation. For example, maybe the GBM tumor segmentation of BRATS and also survival prediction?

**Special Issue:**

yes

---

> ### Author Response · Authors · 2020-03-27
> **Response to Reviewer 4**
>
> We would like to thank reviewer 4 for reading and reviewing our work and providing insightful comments and suggestions.
>
> We would like to clarify and address the following points:
>
> -	“There a few minor typos in the submitted paper.”
>
> Thank you for spotting the errors. We will go though carefully and correct the mistakes in the revised submission.
>
>
> -	“For somebody with no previous knowledge, it is suggested that the authors spend some time and explain what DICOM attributes mean. For example, if Scanning Sequence is known by (0018,0020) -- is this universal and consistent across all sites? or is this vendor specific?”
>
> DICOM attributes are the data elements that are stored in the header of a DICOM image. It is a constant and standardised series of tags that provides information on the scan. The tags are coded as a pair of numbers (0080,0020 etc.) which is the same across all vendors. However, there are different level of implementation and customisation to the entries. For example, tags such as modality, sequences, MRI imaging parameters (TE/TR times) used in the study are generated automatically by the machine and should be very consistent regardless of machines. Study and series descriptions for most medical centres uses vendors default name which normally contains the name of protocols/sequences used for the scan. Certain tags such as Body Part Examined (0018, 0015), although may provide useful information it is an optional entry. Even if entered, the accuracy is dependent of the radiographer's entering it, therefore we did not use them for this study. Thank you for suggestion, we will make it clear in the revised manuscript.
>
> -	“I would also like to see another application implementation. For example, maybe the GBM tumor segmentation of BRATS and also survival prediction?”
>
> Thank you for the suggestion. We also believe that it would be good validation to test the performance transfer learning on classification task or tumour level segmentation. However, due to the virus pandemic in Hong Kong, we are unable to access the GPU workstation to conduct any more experiments. Our model right now, requires very GPU high memory to train with sufficient batch size. This is definitely one area we plan to work on and will release the transfer learning results as soon as possible either in the paper/presentation or the GitHub page.
>
> -	“Have the authors made this trained model publicly available for immediate use?”
>
> Yes, we have updated our GitHub repository to provide the source code of our trained model and the datasets used to train the models.

---

### Meta-Review · Area_Chair1 · 2020-04-08
**MetaReview of Paper114 by AreaChair1**

**Rating:** 4
**Recommendation For Accepted Papers:** Oral

**Metareview:**

Reviewers side on the 'accept' side. There is discussion and the authors are encouraged to update / clarify their paper in accordance.

I recommend to accept.

**Paper Type:**

both

**Special Issue:**

yes

---

### Decision · Program_Chairs · 2020-04-11

Accept